# A Scoping Review of Acute Sedentary Behaviour Studies of People with Spinal Cord Injury

**DOI:** 10.3390/ijerph21101380

**Published:** 2024-10-18

**Authors:** Nathan T. Adams, Bobo Tong, Robert Buren, Matteo Ponzano, Jane Jun, Kathleen A. Martin Ginis

**Affiliations:** 1School of Health and Exercise Sciences, University of British Columbia, Kelowna, BC V1V 1V7, Canada; robert.buren@ubc.ca (R.B.); mponzano@mail.ubc.ca (M.P.); kathleen_martin.ginis@ubc.ca (K.A.M.G.); 2International Collaboration on Repair Discoveries (ICORD), Blusson Spinal Cord Centre (BSCC), University of British Columbia, Vancouver, BC V5Z 1M9, Canada; bobo.tong@ubc.ca; 3Library, University of British Columbia Okanagan, Kelowna, BC V1V 1V7, Canada; jane.jun@ubc.ca; 4Department of Medicine, Division of Physical Medicine & Rehabilitation, University of British Columbia, Vancouver, BC V5Z 1M9, Canada; 5Centre for Chronic Disease Prevention and Management, University of British Columbia, Kelowna, BC V1V 1V7, Canada

**Keywords:** sitting time, cardiometabolic health, psychological health, well-being, inactivity

## Abstract

People with a spinal cord injury (SCI) report less physical activity than other populations and may engage in more sedentary behaviour (SB), especially sitting time. SB negatively impacts physiological and psychosocial outcomes in the general population, yet minimal research has explored the effects in people with SCI. The goal of this scoping review was to catalogue and describe the effects of acute SB among people with SCI. We searched four databases before February 2024 for studies in which people with any SCI sat, laid, or reclined for more than one hour in a day, and any physiological, psychological, or behavioural (i.e., SB time) outcome was measured. In total, 2021 abstracts were screened, and eight studies were included (*n* = 172 participants). The studies were characterized by varied definitions, manipulations, and measures of SB. Most measured outcomes were physiological (e.g., metabolic, blood pressure), followed by behavioural (e.g., SB time) and psychological (e.g., well-being, affect). When SB was interrupted, only postprandial glucose and affect improved. Based on two studies, participants engaged in 1.6 to 12.2 h of SB per day. Average uninterrupted wheelchair sitting bouts lasted 2.3 h. Based on the very limited body of research, it is impossible to draw any conclusions regarding the nature, extent, or impact of SB in people with SCI. There is much work to carry out to define SB, test its effects, and determine if and how people with SCI should reduce and interrupt SB.

## 1. Introduction

People living with a spinal cord injury (SCI) do less physical activity (PA) than people with other chronic conditions [1] and may spend more time engaged in sedentary behaviours (SB) [2]. SB—defined as sitting, lying, or reclining with an energy expenditure ≤ 1.5 metabolic equivalents—is associated with increased cardiovascular disease incidence and all-cause mortality, independent of PA behaviour [3,4,5]. While people with SCI might be presumed to spend more time than the general population in SB, especially sitting, there is very little research evaluating time spent on SB and the consequences in this population.

Guidelines for SB were recently added to the World Health Organization PA guidelines [6] and extended to people living with disabilities [7]. The guidelines suggest people “move more, and sit less” based on many cross-sectional and epidemiological studies, which confirm the negative effects of accumulated SB in the general population [4,8,9]. These initial guidelines lack clarity in regard to when and how SB should be interrupted, making them difficult to follow. There are many interruption strategies shown to mitigate the negative effects of sitting, but the optimal strategies have not been determined [10,11]. Furthermore, extending the general population PA and SB guidelines to people with disabilities has been challenged by scientists because very little of the evidence is based on studies that included people with disabilities [12]. It might be assumed that reducing total SB would have positive effects on the health and well-being of people with SCI, as demonstrated in other populations. However, given the unique physiology and experiences of people with SCI, there is a need to review the relevant existing literature and address knowledge gaps before formulating SB recommendations [13]. Identifying SB-related outcomes could be useful to a population that has an increased risk of cardiovascular disease, metabolic disease, and lower subjective well-being [14,15,16].

Researchers typically study SB with one of two designs: longitudinally, where SB is measured using activity monitors and analyzed in relation to other outcomes, and acutely, where participants are made to sit for a period of time within a single day. In longitudinal studies, greater time spent on SB has been associated with greater cardiovascular disease risk, poorer mental health and well-being, poorer sleep time and quality, and increased risk for mortality [4,8,9,17,18]. Acute studies have shown that a single bout of sitting or SB, without any PA interruption, can result in short-term cardiometabolic burden (acute hypertension, arterial stiffness, glucose, insulin, triglycerides) and reduced cognitive performance [11,19,20,21]. Indeed, greater SB time within one day or accumulated over multiple days has been linked to negative physiological and psychological outcomes in populations without physical disabilities [4,8,9].

In people with SCI, we know much less about SB and its effects. In preparing this review, we confirmed that there are no completed reviews on PROSPERO that discuss SB among people with SCI. Therefore, it is currently not known how much SB people with SCI engage in or how SB impacts overall health and well-being. To begin to explore these questions, we conducted this scoping review to broadly explore and summarize the current published research on SB in people with SCI, identify gaps in the research literature, and propose future research directions for SB among people with SCI.

## 2. Materials and Methods

This scoping review was conducted following the 5-stage methodological framework recommended by Arksey and O’Malley [22], which was updated by Daudt, van Mossel, and Scott [23]. This review is reported in accordance with the Preferred Reporting Items for Systematic Reviews and Meta-analyses extension for Scoping Reviews (PRISMA-SCr) [24]. The review protocol was designed and implemented by a working group consisting of researchers with expertise in SCI, SB, kinesiology, and behaviour change—one of whom has lived experience of SCI—and a medical librarian.

### 2.1. Identifying the Research Question

This scoping review explored the following question: “What is currently known from published, peer-reviewed research about how any quantity of SB impacts people with SCI?”. The Arksey and O’Malley scoping review framework aligns with the objectives of this review, i.e., to examine the extent, range, and nature of research activity, to summarize and disseminate research findings, and to identify research gaps in the existing literature [22].

### 2.2. Data Sources and Searches (Identifying Relevant Studies)

Electronic databases (Embase and Medline, PsycInfo, and CINAHL) were searched from inception by two authors (NA, JJ), one of whom was a library scientist specializing in conducting systematic literature reviews (JJ). We used keywords and MeSH terms related to SCI, SB, prolonged sitting, and expected outcomes, and compiled a list of relevant SB terms and definitions (Table 1). The full search strategy, including Boolean terms, is included in Appendix A. Reference lists of included records were also searched. Reasons for excluding full texts are reported with PRISMA-ScR flow in the results.

### 2.3. Study Inclusion

All citations retrieved by the search were imported to online systematic review software version 2.0. (Covidence, Melbourne, Australia). A minimum of two authors (NA, BT, MP) independently screened titles, abstracts, and full texts in English. Disagreements between the initial screening authors were discussed. If a decision could not be reached, a third author was consulted.

With the expectation that not all studies would use recently published SB definitions and terminology [3], we used broad inclusion criteria to capture as many relevant studies as possible: studies that (i) involved adults of any sex/gender and any distinguishable group of participants who had any SCI, (any lesion type, any lesion level, any ASIA classification) and were at least two weeks post-injury at the time of the study; (ii) measured or manipulated people with SCI sitting, lying, or reclining for more than one hour in a day (as per the definition of SB) [3] using any study design; and measured (iii) any physiological or psychological outcome, or (iv) the amount of sitting time per day. Studies were excluded if they evaluated long-term physical inactivity or a “sedentary lifestyle” (defined as not meeting PA guidelines) [3] or if the study involved a PA behaviour-change intervention and the interventionists mentioned that in trying to increase PA, they were also trying to reduce SB as a secondary outcome. To capture relevant studies that explored the effects of acute SB in lab-based settings, a minimum SB bout of 60 min was required. This criterion aligns with the recommendation that to enhance the ecological validity of prolonged sitting research, sitting protocols should be 1–2 h in duration [21], noting that most SB research measures effects within or beyond this duration [26].

### 2.4. Data Extraction (Data Charting)

We extracted the following information from each study included in the review: author, publication year, participant demographics (age, sex, etc.), details of interventions, (study purpose, study design, primary question/comparison, how SB was defined or manipulated), and all outcomes. Tables with all of the extracted information are included in Appendix A [22]. Once all outcomes were extracted, they were categorized as either physiological, psychological, or behavioural (e.g., quantification of SB time).

Data extraction was completed independently by two authors (NA and BT) using Covidence online systematic review software, which automatically compared the agreement between the two authors. If a consensus was not reached, authors reviewed the articles together to reach agreement on which value to use. Articles were not blinded during data extraction and charting. Additional tables with information on the study findings, their definitions of SB, and full data summaries are included in Appendix A.

### 2.5. Summarizing the Results

Information from each study is summarized in Table 2 and Table 3. The results are organized around the three categories of outcomes (physiological, psychological, behavioural).

## 3. Results

### 3.1. Literature Search and Study Selection

After removal of 741 duplicate records, a total of 2021 potentially eligible articles were identified. Following screening of abstracts and titles, 1941 were excluded because they did not meet inclusion criteria. Eighty records remained and their full texts were retrieved for screening. Of these full-text records, eight studies were included in the review (Figure 1).

### 3.2. Description of the Included Trials

#### 3.2.1. Trial Setting and Participants

Study characteristics are summarized in Table 3. One article was published in 1988 and all other articles were published between 2006 and 2020. The majority of studies were conducted in the United States (*n* = 3), with others occurring in the Netherlands (*n* = 2), the United Kingdom (*n* = 2), and Japan (*n* = 1).

The number of participants in each study ranged from 5 to 47 (median = 16), with a total of 172 participants across all studies. Four studies included only male participants. The mean age of the participants ranged from 32.7 years [28] to 54.5 years [31]. Time since injury was not reported in six of the eight studies. The remaining studies included participants at an average of 89.6 days [31] and 12.3 years post-injury [28]. Participants varied in their level and completeness of spinal cord injury, as well as the method of reporting among the eight studies (Table 2).

#### 3.2.2. Sedentary Behaviour Definitions, Study Designs, and Study Contexts

Information about the definition or manipulation of SB within each study is included in Table 4. Four study designs were used: randomized crossover trial (*n* = 2), quasi-experimental (*n* = 3), cross-sectional (*n* = 2), and longitudinal (*n* = 1). In five studies, SB was induced in a lab-based setting. In three studies, it was observationally measured in free-living contexts.

#### 3.2.3. Outcomes Measured

Within three categories of outcome measures (physiological, psychological, behavioural), the number of unique outcome measures and number of studies that included them are presented in Table 5 and Appendix A. Physiological outcomes were metabolic, blood pressure, fluid shifts, and heart rate. Psychological outcomes were well-being, affect, exercise self-efficacy, and fatigue. Behavioural outcomes were sedentary time, seated time, pressure relief behaviour, motility, and PA. With the exception of the longitudinal study, all outcomes were measured acutely during SB.

### 3.3. Summary of Key Findings

#### 3.3.1. Physiological Outcomes

One study compared a sedentary condition to an interrupted sedentary condition and found no significant differences in most metabolic and blood pressure outcomes, but a significant improvement in one metabolic outcome—postprandial glucose—in the interrupted condition (Blood glucose iAUC: interrupted, 1.9 mmol/L × 2.5 h [1.0, 2.7], prolonged sitting, 3.0 mmol/L × 2.5 h [2.1, 3.9], *p* = 0.015; Blood glucose tAUC: interrupted, 15.3 mmol/L × 2.5 h [14.4, 16.1], prolonged sitting, 16.4 mmol/L × 2.5 h [15.5, 17.2], *p* = 0.015) [27].

Fluid shifts, clearance of water, and other blood aspects of kidney function were measured in three of the lab-based studies [28,29,34]. After recovery from exercise, none of the renal outcomes changed over a two-hour sitting period [28]. Over an hour of sitting and an hour of recumbency, there were seated and recumbent differences in urinary sodium in people with paraplegia (seated, 19.7 mmol ± 6.9; recumbent, 10.3 mmol ± 3.2) and differences in urinary potassium for both people with paraplegia (seated, 14.1 mmol ± 6.1; recumbent, 6.5 mmol ± 1.9) [29]. In the seated posture, the group with paraplegia and the group with tetraplegia has significantly different urine volumes (paraplegia, 476 mL ± 77; tetraplegia, 401 mL ± 88), urinary sodium (paraplegia, 19.7 mmol ± 6.9; tetraplegia, 3.9 mmol ± 1.3), urinary potassium (paraplegia, 14.1 mmol ± 6.1; tetraplegia, mmol ± 0.9), and urine osmolality (paraplegia, 212 mosm ± 61; tetraplegia, 121 mosm ± 21). Lastly, the infusion of L-NAME to inhibit nitric oxide resulted in significant changes to plasma renin (−2.6 pg/mL ± 5.6) and aldosterone (−19.0 pg/mL ± 46.1), which is likely associated with higher blood pressure induced by the L-NAME during an hour of sitting [34].

Two studies measured autonomic control of heart rate. In the first study, responses to two hours of sitting after exercise were significantly different between fit and unfit people with SCI; fit people had greater cardiac autonomic control in a recovery period after exercise than unfit people (group main effect, *p* < 0.01) [33]. In the second study, when participants received L-NAME, their blood pressure was increased and normalized compared to when they received a placebo during an hour of sitting (MAP, +17.3 mmHg ± 10.3) [34].

#### 3.3.2. Psychological Outcomes

Psychological outcomes were measured in two studies: one that compared a sitting condition to an interrupted sitting condition [27], and one that measured outcomes over an observational period of four days [30]. Comparing the two sitting conditions, well-being, fatigue, and exercise self-efficacy did not change, but positive affect improved in the interrupted sitting condition (positive affect, +4.8, *p* = 0.001; negative affect −1.1, *p* = 0.079) [27]. Exercise self-efficacy was not related to SB during the 4-day observational study (p^c^ = −0.06, *p* = 0.73) [30].

#### 3.3.3. Behavioural Outcomes

Sitting was measured in one study and SB in two studies [30,31,32]. In the sitting study, pressure sensors were used to collect data [32]. The average total time spent sitting in a wheelchair was 10.6 h per day. Of note, the pressure sensors could not detect upper body movement or wheelchair propulsion, which may have interrupted SB, making this measure sitting time, not SB. On average, participants did 0.4 pressure reliefs and 2.4 weight shifts per hour of wheelchair sitting. An average wheelchair sitting bout lasted 2.3 h without a pressure relief or weight shift, although it is unclear if this could have included wheeling, which may have interrupted this sitting bout by increasing energy expenditure.

One study used accelerometers to measure SB and reported an average of 1.55 h per day of SB, based on four consecutive days of data collection [30]. The other SB study also used accelerometry and measured SB time per day at three time points after discharge from in-patient rehabilitation [31]. The study reported a significant decrease in SB from discharge (12.2 h/day) to six months (11.0 h/day; −1.1 h/day, *p* < 0.001) and 12 months post-discharge (11.1 h/day; −1.0 h/day, *p* < 0.001). When SB time was counted only if it lasted at least 30 min, prolonged SB did not change over the year (discharge, 7.6 h/day; 6 months, 7.1 h/day; 12 months, 7.0 h/day). Over a whole year of observation, SB time was associated with ambulation ability (β = −108.5 min/day [−155.6, −61.3], *p* < 0.001) but not the time since injury (β = 1.8, *p* = 0.059) nor age (β = 0.6, *p* = 0.92).

## 4. Discussion

This scoping review was conducted to explore and summarize the current published research on SB in people with SCI, identify gaps in the research literature, and propose future research directions for SB among people with SCI. Through a systematic search, eight articles were identified that met the review’s inclusion criteria. The articles quantified the amount of SB that people with SCI engage in and explored SB, physiological outcomes, and psychological outcomes. There is vast heterogeneity among these studies in terms of the research questions, the interventions tested, and both the number and type of outcomes assessed and measures used. The following discussion summarizes the research, highlights knowledge gaps, and provides recommendations for future research.

### 4.1. Behavioural Outcomes

We found three studies that measured wheelchair-sitting time or SB [30,31,32]. The study of sitting reported that participants sat in their wheelchairs for nearly 11 h per day, on average [32]. It is important to note that sitting is not the same as SB and the study was not designed to measure energy expenditure or upper body movement such as wheelchair propulsion (i.e., behaviours that would help distinguish sitting time from sedentary time). On the other hand, sitting time was only measured in the wheelchair; participants may have transferred and sat in other chairs at other times in the day. Nevertheless, the results are informative in showing that people with SCI spend a significant portion of the day seated in their wheelchair, some of which could be assumed to be a proxy for SB. If combined with other techniques for measuring movement (e.g., accelerometery), the pressure sensor methodology used to measure sitting in this study could be applied to measure SB in future studies.

Two studies used accelerometery to measure SB and reported estimates from 1.5 h per day to 12.2 h per day [30,31]. This is a big difference that reflects differences in how SB was calculated in each study. In the first study, periods of SB were counted only if they lasted longer than 30 min and just 5 s of movement were counted as an interruption of SB time [30]. Thus, very brief and light-intensity movements such as gesturing and eating stopped the counting of sedentary time. Given that 1.5 h/day of SB is much less than estimates of SB time in the general population [35,36,37], Nooijen et al.’s results are likely a profound underestimation of actual SB time among people with SCI. In contrast, Postma et al. calculated prolonged SB such that a prolonged bout of SB was considered to end when the person was active for a full minute. Their report of 12.2 h per day of SB is 156% greater than estimates of SB in the general population (7.7 h SB/day) and is likely a more accurate estimate of SB in people with SCI [38].

Based on studies conducted among people without disabilities, >9.5 h of sedentary time significantly increases the risk of mortality [4,5]. It is well documented that living with an SCI increases the risk of heart disease and stroke [14], and cardiovascular diseases account for significant mortality in people with SCI [39]. Several factors explain the increased risk of chronic disease and mortality post-SCI such as structural changes to the cardiovascular system, autonomic dysfunction, and reduced PA [1]. As in the general population, SB may be an independent risk factor for disease and mortality [4]. We urge scientists to conduct epidemiological studies that collect both PA and SB data from people with SCI and test these behaviours as independent risk factors for morbidity and mortality. When conducting such studies, it will be important to use clear definitions and operationalizations of SB and SB interruptions that are appropriate for people with SCI.

### 4.2. Cardiovascular Outcomes

Five studies measured various cardiovascular and autonomic outcomes including blood pressure, heart rate, autonomic control, and vagal outcomes [27,28,29,33,34]. In the three studies that tested perturbations or interruptions, neither arm crank ergometry nor changes in posture had effects on cardiovascular outcomes; however, the administration of L-NAME maintained or increased blood pressure during sitting. While these studies do not explain how SB affects cardiovascular health among people with SCI, given the greater risk of cardiovascular disease among people with SCI compared to the general population [14], going forward, it will be important to determine how interrupting SB could improve various cardiovascular outcomes.

Blood pressure was measured in four of the studies [27,28,29,34]. The arm crank ergometry interruption condition did not change blood pressure compared to a prolonged sitting condition [27]. Many studies that interrupt prolonged sitting have found reductions in blood pressure relative to prolonged sitting when interruption stimuli are of sufficient intensity [19]. Either this interruption was not intense enough for a response, or other aspects of SCI physiology are unique, and BP may not be affected by the prolonged sitting control condition in the same way. Two studies found significant differences in blood pressure in response to changes in posture (sitting vs. recumbency) [29] and nitric oxide inhibition (L-NAME injection vs. placebo injection) [34]. In the general population, one hypothesis connecting SB to cardiovascular disease posits that SB induces lower-limb blood pooling and acute hypertension, [40] while the inability to induce vasoconstriction among people with SCI [41,42,43] could result in maintained or lower BP during prolonged sitting, especially in those with higher level injuries. Further studies need to explore the specific BP response to prolonged sitting among people with SCI.

Heart rate was measured in two studies, both of which found significant changes when altering posture and injecting L-NAME, respectively [29,34]. Within these two studies, the changes in heart rate are likely driven by increased blood pressure, allowing for regulation of cardiac output with lower heart rate. Heart rate control is unique and heterogeneous among people with SCI, but it is not clear how responses to SB may change for people with higher or lower injuries.

One study measured autonomic control, vagal recovery, sympathovagal balance, and the recovery of HR during a SB recovery period following exercise [33]. These studies found differences in autonomic control between fit and unfit people with SCI adding to the heterogeneity of responses, as it is unclear if fitness can impact responses to SB [44,45]. Improved autonomic control in some individuals may explain their improved fitness and ability to respond favourably to both PA and SB.

The cardiovascular outcomes discussed above have both acute [11,19] and chronic [4,5] associations with SB in the general population. Cardiovascular outcomes and the potential impact of SB are important for people with SCI since they are at a greater risk of stroke and heart disease than other populations [14], and the rate of cardiovascular mortality is high among those living with SCI [39]. Since reductions in SB improves the risk profile for other populations, future research needs to determine if the same negative associations exist between SB and risk in people with SCI and if they can be mitigated by interrupting SB.

### 4.3. Metabolic Outcomes

During the one study where arm crank ergometry interrupted sitting, only post-prandial glucose was lowered [27]. It is surprising that only one study measured metabolic outcomes since people with SCI have 1.66 times greater odds of being diagnosed with type 2 diabetes when adjusting for age and sex. People with SCI also have a 2.45 times greater chance of type 2 diabetes when adjusting for age, sex, smoking status, hypertension status, body mass index, PA, alcohol intake, and dietary behaviours [15]. In the general population, greater SB time is associated with type 2 diabetes incidence and risk of metabolic syndrome [46,47]. Since metabolic improvements have been found in response to interruptions in the general population, future studies should investigate how people with SCI could improve both acute (post-prandial responses) and chronic (fasting insulin, insulin resistance, and HbA_1_C) metabolic health [48,49,50].

### 4.4. Renal Outcomes

The effects of SB on renal function are difficult to interpret because none of the studies compared between a sedentary and a physical activity condition. One study that compared two sedentary postures found differences in urinary sodium, osmolal clearance, and urinary potassium, suggesting plasma volume shifts and redistribution of blood flow during seated and recumbent postures may be different [29]. Kidney function and the movement of fluids during SB are important to understand as SB may influence the incidence of autonomic dysreflexia, neurogenic bowel, or chronic kidney disease. It has been suggested that disruptions to kidney function may at least partially explain the increased risk for cardiovascular complications associated with SB, as increased lower limb blood pooling would reduce kidney pressure, activating renin-angiotensin pathways in an effort to increase blood pressure [40]. With poorer redistribution of blood flow, people with SCI may have a greater blood pooling effect and, therefore, greater activation of the renin-angiotensin to increase blood pressure, leading to stress on the system. However, it is not yet clear how the kidneys of people with SCI respond to longer bouts of SB. Future research should measure SB more directly in relation to fluid shifts and kidney function and compare it with control conditions to better understand its relationship to autonomic dysreflexia or chronic kidney disease, specifically for people with SCI.

### 4.5. Psychological Outcomes

Only two studies measured psychological outcomes [27,30]. Overall, there were no differences in psychological outcomes in a study that compared uninterrupted versus interrupted SB conditions, with the exception of more positive affect in the interrupted condition [27]. Both studies measured exercise self-efficacy and found it to be unrelated to SB [27,30]. With regard to the null findings, when comparing psychological outcomes across two SB conditions, some aspects of psychological well-being (e.g., subjective well-being) may be too stable to be influenced by changes in acute SB. Likewise, small and short SB interruptions may not be enough to improve complex psychological states and traits. For example, Bailey et al. [27] included a measure of well-being that included items such as “Overall, how satisfied are you with your life nowadays?”, “Overall, to what extent do you feel the things you do in your life are worthwhile?”, and “I’ve been feeling loved” [51,52]. It is unreasonable to expect scores on these items to change over the course of a single SB interruption over a 5 h period. Greater attention needs to be paid to the selection of psychological outcome measures in SB studies, with thought given to whether it is realistic to expect changes in particular outcomes and the mechanisms by which those changes may occur.

Both studies used Schwarzer and Renner’s Physical Exercise Self-Efficacy Scale and found no significant changes or outcomes related to SB. This null finding likely reflects the task-specific nature of self-efficacy [53] and this questionnaire’s focus on self-efficacy for exercise rather than self-efficacy related to SB or engaging in interruptions. A brief SB interruption is unlikely to make a person with SCI feel more confident in their ability to participate in exercise. Future SB studies may investigate self-efficacy but should specifically choose to ask about self-efficacy for interrupting SB.

Notably, very few psychological measurements were included in the reviewed studies. Psychological outcomes were the smallest outcome category, with only five measures across all studies. Future acute SB studies could benefit from including more measures of mood, activation, self-perceptions (e.g., body image) and psychophysiological outcomes, including symptoms of fatigue and pain. Studies of chronic SB would benefit from including measures of symptoms of depression and anxiety and aspects of health-related quality of life. These outcomes are related to SB in the general population [9], yet have not been explored in people with SCI.

### 4.6. Limitations and Strengths

This scoping review has numerous strengths, including the rigorous systematic methods used for identifying and evaluating research evidence in accordance with pre-developed scoping review guidelines [22]. Perhaps the biggest limitation is that no single outcome was evaluated in more than four studies, making it difficult to identify clear patterns for any outcome. Additionally, the samples were broad and represented the heterogeneous population of people with SCI, as is typical of human participants research among the SCI population. However, this review does identify where research is needed and potential outcomes to explore in future studies. Another limitation is that by design, scoping reviews do not incorporate an evaluation of study quality since the goal of scoping reviews is to determine the scope of research rather than answer a clinically meaningful question or provide evidence to inform clinical practice [54]. Therefore, the quality of the reviewed studies was not appraised and we cannot make statements about the quality of current SB research among SCI samples. Nevertheless, by including all eligible studies, regardless of quality, we were able to fully represent the current state of science related to SB among people with SCI [22].

Another significant limitation is that while four of the eight included studies met our inclusion criteria, they were not explicitly designed as studies that manipulated or measured SB. These four studies were conducted in lab-based settings and fit the inclusion criteria used in other SB reviews of >1 h of uninterrupted sitting [19]. While not explicitly designed to measure SB or prolonged sitting, these studies give information about physiological phenomena that happen while people with SCI engage in SB. We opted to include these studies to maximize the number of studies in our review. We recognize that other researchers may have elected to exclude these studies. Going forward, researchers’ use of clear and consistent definitions of SB and including SB as the primary focus of research would facilitate consistency in interpreting study findings. Also, we chose not to include gray literature within this review. With the goal of identifying and surveying the current published research related to sedentary behaviour and to better understand how scientists are addressing this issue, gray literature would not have helped achieve that aim.

Likewise, there are some studies that we did not include because they did not meet our inclusion criteria (i.e., did not measure or manipulate SB), but the results point toward other outcomes that may be linked with SB. Specifically, a couple of studies that looked at wheelchair modifications suggested a potential connection between wheelchair seating configurations and pressure injuries, which are a significant problem among people with SCI [55,56]. In addition, in a pain survey study, some participants reported that SB may exacerbate their neuropathic pain [57]. None of these studies measures or manipulated SB, but their outcomes suggest both pressure injuries and neuropathic pain should be investigated in SB studies.

Scientific definitions of SB have only recently been developed, and we believe that the current language used to describe SB is ableist [58]. People with SCI may not be able to interrupt their sitting with a posture change, depending on the level and completeness of their injury. Those who cannot change posture or increase energy expenditure could view this language as inequitable and ableist. This ableist language represents a limitation to the adoption and translation of SB research. People with SCI should be engaged in efforts to adapt SB and prolonged sitting language to be more inclusive and respectful to all individuals, regardless of ability. Sitting is a complex behaviour to change in terms of the contexts (e.g., home, work) and how it can interact with other behaviours (e.g., eating, screen time), especially for individuals who use wheelchairs. The language used in this review to describe SB was intentional, as it represents the language commonly used in the SB literature. However, the authors considered alternative labels such as “passive sitting” or “stationary behaviour” to represent the unique experience of people with SCI who may not be able to easily interrupt “sitting”. Since the scientific body of research on SB and people with SCI is still small, we are in an opportune moment to engage people with SCI first-hand to determine preferences for discussing SB and prolonged sitting moving forward.

## 5. Conclusions

Although SB has been well studied in the general population as an independent risk factor for morbidity and mortality, we identified only eight studies that measured SB or its outcomes in people with SCI. The studies were characterized by varied definitions, manipulations and measures of SB, and a limited range of outcome measures that cut across physiological, psychological, and behavioural outcomes. Based on the very limited body of literature, it is impossible to draw any conclusions regarding the nature, extent, or impact of SB in people with SCI. There is much work to carry out to define SB, test its effects, and determine if and how people with SCI should reduce and interrupt SB.

## Figures and Tables

**Figure 1 ijerph-21-01380-f001:**
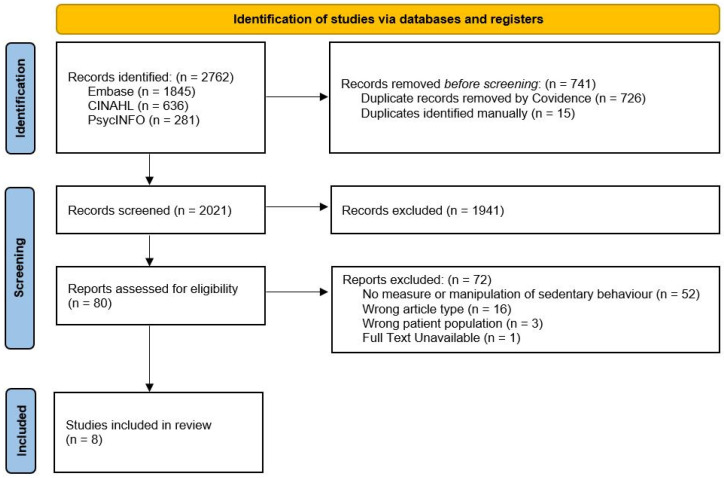
PRISMA flow diagram of study inclusion and exclusion.

**Table 1 ijerph-21-01380-t001:** Relevant sedentary behaviour terms and definitions used to inform the search strategy.

Terms	Definitions
Sedentary Behaviour	“Sedentary behavior is any waking behavior characterized by an energy expenditure ≤ 1.5 metabolic equivalents (METs), while in a sitting, reclining or lying posture” [3].
Sedentary Behaviour Bouts	There is no standard operational definition for how long sedentary behaviour must last to be a “bout”. Based on ecological validity and behavioural studies, people regularly engage in sedentary behaviour for at least an hour at a time multiple times a day [25]. We defined a bout as sedentary behaviour that lasted at least 60 min without interruption.
Prolonged Sitting	There is no standard operational definition of “prolonged” sitting. The majority of “prolonged sitting” laboratory-based studies have individuals sit for at least one hour. We defined it as sitting for at least 60 min without interruption.
Sedentary BehaviourInterruption	Within a sedentary behaviour bout, an “interruption” typically refers to some kind of light physical activity behaviour that either changes posture from the defined sedentary postures or increases energy expenditure > 1.5 METs. We adopted this definition.
Passive Sitting	“Passive sitting refers to any waking activity in a sitting posture characterized by an energy expenditure ≤ 1.5 METs” [3].
Active Sitting	“Active sitting refers to any waking activity in a sitting posture characterized by an energy expenditure > 1.5 METs” [3].

**Table 2 ijerph-21-01380-t002:** Participant characteristics included within each study.

Study ID	Bailey [27]	Kawasaki [28]	Kooner [29]	Nooijen [30]	Postma [31]	Sonenblum [32]	Wecht [33]	Wecht [34]
	2020	2012	1988	2015	2020	2016	2006	2009
Total *n*	14	14	12	37	47	25	18	5
Lumbar	5							
Thoracic	8					23		
Cervical						2		
Post-Polio	1							
Paraplegic	14	14	6	25	24		18	
Tetraplegic			6	22	23			
Incomplete	10			13				
Complete	4	14	12	24				
ASIA A		14						1
ASIA B								4
ASIA C					4			
ASIA D					43			

**Table 3 ijerph-21-01380-t003:** Characteristics of included studies.

Study	Year	Sample (*n*)	Female (*n*)	Female (*n*, %)	Age (Mean, SD ^1^)	Study Purpose	Study Design	Outcome Measures ^2^	Summary of Findings
Bailey [27]	2020	14	8	57.14	51	To compare effects of uninterrupted sitting to interrupted sitting, where every 20 min participants completed 2 min of arm crank ergometry.	Randomized-Crossover Trial	Physiological, Psychological	Compared to uninterrupted sitting, interrupted sitting over 5.5 h did not improve physiological outcomes (BP, glucose, insulin, triglycerides) but did improve some psychological outcomes (positive and negative affect).
Kawasaki [28]	2012	11	0	0.00	32.7	To investigate the renal and endocrine response, and recovery from arm exercise among people with spinal cord injury compared to an able-bodied sample.	Quasi-Experimental	Physiological	People with spinal cord injury had attenuated increase in aldosterone and adrenaline during a two-hour sitting bout following exercise indicating sympathetic dysfunction.
Kooner [29]	1988	12	0	0.00	NR ^3^	To compare the hemodynamic, hormonal, and urinary effects of postural change from sitting to recumbency in individuals with tetraplegia, paraplegia, and individuals with able bodies.	Quasi-Experimental	Physiological	People with tetraplegia had lower sitting blood pressure during an hour of sitting, which significantly increased when transitioning to an hour recumbency. People with spinal cord injury had higher renin and aldosterone activity than people without spinal cord injury, and urine output was lower in people with tetraplegia. Postural change may be a method for reducing orthostatic hypotension in people with spinal cord injury that occurs during prolonged sitting.
Nooijen [30]	2015	37	5	13.51	44 ^4^	Using accelerometry, to explore associations of physical activity, sedentary behaviour, and motility with exercise self-efficacy.	Observational Associations, part of a larger Randomized Control Trial	Behavioural, Psychological	Sedentary behaviour was not associated with exercise self-efficacy. Accelerometry measured sedentary behaviour occurred 1.55 h per day measured over a four-day period.
Postma [31]	2020	47	22	46.81	54.5 (12.9)	To evaluate changes in physical activity and sedentary behaviour from discharge to one year after inpatient rehabilitation in ambulatory individuals with spinal cord injury using accelerometry.	Longitudinal Cohort Study	Behavioural	Sedentary behaviour decreased in the first year following rehabilitation (−1.0 h/day, *p* < 0.001), and the highest sedentary time across all measurement points was 12.2 h per day.
Sonenblum [32]	2016	28	6	21.43	41 (12)	Using chair pressure sensors, to describe in-seat movement including weight shift and pressure relief behaviours compared to clinical guidelines.	Multi-day Observational Study	Behavioural	People with spinal cord injury spent 10.4 h a day sedentary in their chair measured over 1–2 weeks per participant. On average, there was a 2.3 h sitting bout where no pressure relief, or weight shift occurred.
Wecht [33]	2006	18	0	0.00	39 (7.5)	To compare the seated response to exercise in fit and unfit people with paraplegia. Fit people were identified as those participating in regular aerobic training, and able to match 85% of their predicted VO_2_.	Quasi-Experimental	Physiological	Fit people with paraplegia had greater ability to recover autonomic function during a two-hour sitting bout after exercise, and improved cardiac autonomic control compared to unfit people with paraplegia.
Wecht [34]	2009	5	0	0.00	40 (10)	To observe the effects of nitro-L-argnine methyl ester (L-NAME) on vascular outcomes in people with tetraplegia and people without physical disability.	Randomized-Crossover Trial	Physiological	Administering L-NAME maintained blood pressure compared to a placebo infusion during one-hour of sitting among individuals with tetraplegia, suggesting that L-NAME is a potential method for avoiding orthostatic hypotension.

^1^ SD was not reported in all studies. ^2^ Outcome measures were categorized as physiological, psychological, or behavioural and listed in order depending on which appeared first within an individual study. ^3^ NR = not reported. ^4^ Age reported in Nooijen et al. 2015 [30] as median.

**Table 4 ijerph-21-01380-t004:** How sedentary behaviour was measured or manipulated within each included study.

Study	Measure or Manipulation of Sedentary Behaviour
Bailey [27]	5.5-h uninterrupted sitting bout, and an equal bout with physical activity interruptions every 30 min.
Kawasaki [28]	2 h sitting bout as a recovery from exercise.
Kooner [29]	1 h of sitting and 1 h of lab-based recumbency.
Nooijen [30]	Accelerometry measured daytime sedentary bouts, defined as 30 min or more, without a physical activity interruption of 5 s.
Postma [31]	Accelerometry measured body postures and movements, automatically classified by the accelerometer device. Accumulated sitting and lying made up the duration of sedentary behaviour.
Sonenblum [32]	Chair pressure-relief sensor measured sitting time.
Wecht [33]	1.5 h sitting bout as a recovery from exercise.
Wecht [34]	1 h supine bout, and 0.5 h semi-recumbent session after a progressive head-up tilt.

**Table 5 ijerph-21-01380-t005:** Summary of included measures.

Constructs or Categories	Measurement	Number of Studies	Number of Measures
Physiological	Blood Pressure	4	3
	Fluid Shifts	3	10
	Heart Rate and Recovery	3	5
	Metabolic Outcomes	1	6
Psychological	Exercise Self-Efficacy	2	1
	Well-Being	1	2
	Affect	1	1
	Fatigue	1	1
Behavioural	Sedentary Time/Day	2	3
	Pressure Relief Behaviour	1	3
	Motility and Physical Activity	1	2
	Wheelchair Seated Time	1	1

Number of studies refers to studies that measured a particular outcome. Number of measures refers to the number of different assessment techniques used across all studies.

## Data Availability

The original contributions presented in this study are included in the article/Appendix A; further inquiries can be directed to the corresponding author.

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
