# Peer review of "A Scoping Review of Acute Sedentary Behaviour Studies of People with Spinal Cord Injury"

_ijerph, 2024, doi:10.3390/ijerph21101380_

Round 1
Reviewer 1 Report
Comments and Suggestions for Authors
This review article has investigated an interesting and important issue, that is, the investigation of sedentary behavior in people with spinal cord injury.
As the authors mentioned about the limitations of the study, there are indeed many limitations. Although it has been pointed out correctly and it does not reduce the importance and quality of the study, but maybe some things can be changed.
- According to the criteria for study inclusion, it is considered very strict. Couldn't the authors have included more studies and then categorized the studies based on their findings?
- Variables such as pain, quality of life and occurrence of bedsores are of great importance in this group of people. Why did the authors ignore these indicators??
- In the article, the evaluation of the methodological quality of studies not observed. Did the authors evaluate the final 8 studies in terms of research quality?
Despite these comments and as the authors stated, this study provides new knowledge and quality for future studies in this field.
Reviewer 2 Report
Comments and Suggestions for Authors
It's an interesting study. I have only minor suggestions for your consideration;
Line 36, for definition of SB, energy expenditure of SB should be corrected to <=1.5 METs
Line 106, for inclusion criteria, it included only adults, but in the supplement 1, it says adults and children
Line 125, the supplementary files should be supplement 2 to be exact.
Table 4, the order of the categories (e.g. physiological) should be run according to the order that they appear in the texts.
Supplement 1, for types of study, why this review included reviews? as reviews are not supposed to be included in a systematic/scoping review, unless it is a review of reviews.
Reviewer 3 Report
Comments and Suggestions for Authors
Thank you for the opportunity to review this manuscript reporting on a scoping review of literature investigating the effects of sedentary behavior (SB) on people with spinal cord injury (SCI). This contribution is novel because most available scoping and systematic reviews on this topic focus on the effect of physical activity behavior in individuals with SCI, as opposed to SB (the converse). The manuscript is well laid out, utilizes the appropriate reporting guidelines for a scoping review, and demonstrates a comprehensive search and interpretation of the 8 appropriate studies including in the review.
Additional details describing relevance of SB (definition, measurement and assessment) to individuals with SCI recommended in the introduction, discussion, and conclusion. The authors point out that physical activity guidelines for adults with disabilities may not be appropriate for individuals with SCI given that many of the studies leading to that recommendation did not include individuals with this lived experience. Relatedly, it is unclear how SB should be defined and appropriately measured for this population. Based on current definitions of SB and as explained in the manuscript, individuals with SCI may not be able to change their SB depending on their level of injury and wheelchair user status. It is acceptable that the included studies demonstrate heterogeneity in SCI ASIA classification and spinal levels of injury, but this should be emphasized in the discussion as a point for future development in SB research.
The authors begin to acknowledge this issue by explaining current language to describe SB is ableist (discussion). This section should be expanded to include the above consideration.
Reviewer 4 Report
Comments and Suggestions for Authors
Thank you for you’re the opportunity to revise the manuscript titled: A Scoping Review of Acute Sedentary Behaviour Studies of People with Spinal Cord Injury. I would like to commend the authors on the quality of the reporting and the methodology utilised. The topic is of great significance due to the lack of substantial evidence in the area. The manuscript aims to shed a light on the impact of sedentary behaviour in people with SCI and provides a good starting point from where more research should be drawn. I have suggested some minor clarification points below.
Abstract: Well-written and nicely structure. Can the authors please clarify in line 26, what do you mean by wheelchair sitting bouts? Were those while resting (not moving/ pushing the wheelchair)?
Introduction: Describes the problem and relevance of the study well. My only suggestion is to add a bit more detail on the cardiometabolic changes (what are they?) identified in the previous literature in the general population that are a result of SB and how those changes are often more prevalent and more dangerous for people with SCI.
Methods: Well-described. Just a couple of things to add: were the databases searched from inception or was there an initial date/year limit? Were only articles in English included? And if so, provide a justification. Lastly, why grey literature was not searched? Please provide a justification.
Results: Clearly presented.
- Please consider including the overall injury characteristics (time since injury, severity (complete/incomplete and tetra x para) of participants in the included studies.
- Consider including in the table 2 ( for all studies) and in section 3.2.3 (on average) the timeframe that the outcome measures were taken in each study. Was it immediately after SB was observed/ manipulated?
- Also in table 2, consider adding some numerical results where possible.
- Table 2:
o Bailey – how much the psychological outcomes improved between groups?
o Postma – include under summary of findings how much SB reduced in the first year.
o Wretch 2009 – what was the criteria used to define fit participants?
o Wretch 2006 – it is unclear how the study analysed SB. Please clarify. You do clarify it in the discussion, but it would be good to have some information in the results.
- Prisma flowchart: You mention in the methods that all study designs were included, however, in the PRISMA flowchart, 16 articles were excluded due to wrong article type and one due to wrong study design. Can you please clarify?
- Sections 3.3.1 and 3.3.2 – please add some numerical results of studies when possible:
o i.e. line 177: a significant improvement in one metabolic outcome - how much was the improvement?
o Lines 178-192 – clarify what were the differences (numerical) between the groups or conditions, when mention significant changes.
o Table 4. What was the timeframe or interval when the measurements were taken in each study?
Discussion: The discussion is well-written and provided insightful interpretation of the results and recommendations for future research. It can be used as a guide for researchers in the area to progress the evidence in the area. My only suggestion is to discuss in the strengths and limitations sections why grey literature was excluded and potential impacts of it.
